# Rapid Preparation of Novel Ionic Polymer–Metal Composite for Improving Humidity Sensing Effect

**DOI:** 10.3390/polym15030733

**Published:** 2023-01-31

**Authors:** Chun Zhao, Yujun Ji, Gangqiang Tang, Xin Zhao, Dong Mei, Jie Ru, Denglin Zhu, Yanjie Wang

**Affiliations:** Jiangsu Provincial Key Laboratory of Special Robot Technology, Hohai University, Changzhou Campus, Changzhou 213022, China

**Keywords:** IPMC, electrode, Ag NWs, high strength, humidity sensing

## Abstract

Ionic polymer–metal composites (IPMCs) have attracted attention in recent years due to their integration of actuation and sensing functions. As one of the main sensing functions of IPMCs, humidity sensing has been of consistent interest in wearable health monitors and artificial skin. However, there are still some technical challenges in that classical IPMCs have poor humidity sensing performance due to their dense surface electrode, and IPMCs are damaged easily due to an electrode/membrane mismatch. In this work, through the spraying and electrodepositing process, we developed an efficient method to rapidly prepare a Au-shell-Ag-NW (silver nanowire)-based IPMC with high strength, low surface resistance and excellent humidity sensing performance. Meanwhile, we optimized the preparation method by clarifying the influence of solvent type and electrodepositing time on the performance of the Au-shell-Ag-NW-based IPMC, thus effectively improving the humidity sensing effect and strength of the IPMC. Compared with previous research, the humidity electrical response (~9.6 mV) of the Au-shell-Ag-NW-based IPMC is at least two orders of magnitude higher than that of the classical IPMC (~0.41 mV), which is mainly attributed to the sparse gap structure for promoting the exchange of water molecules in the environment and Nafion membrane, a low surface resistance (~3.4 Ohm/sq) for transmitting the signal, and a seamless connection between the electrode and Nafion membrane for fully collecting the ion charges in the Nafion membrane. Additionally, the Au-shell-Ag-NW-based IPMC could effectively monitor the human breathing process, and the humidity sensing performance did not change after being exposed to the air for 4 weeks, which further indicates that the Au-shell-Ag-NW-based IPMC has good application potential due to its efficient preparation technology, high stability and good reproducibility.

## 1. Introduction

Ionic polymer–metal composites (IPMCs) have a typical sandwich structure composed of a Nafion membrane and two flexible electrodes, which have attracted extensive attention in micro-electrical mechanical systems and intelligent sensors in the past decades due to their excellent actuation and sensing functions [1,2,3]. In fact, movable cations and solvent molecules in an IPMC are forced to migrate inversely along the thickness under an electric field, resulting in macroscopic bending deformation. Conversely, an IPMC bends macroscopically under an external mechanical force and generate a strain gradient inside the Nafion membrane, which causes a physical phenomenon in that cations and solvent molecules migrate and accumulate along the strain gradient, generating electrical signals between both sides of the IPMC. As one kind of ionomer, Nafion polymer exhibits a special solid–liquid two-phase structure, including polymer long chains and ionic electrolytes. A polymer long chain has good affinity for water molecules due to it being a hydrophilic sulfonic acid group. Therefore, the change in humidity leads to the local anisotropic distribution of ions and water molecules in the IPMC. Generally, under a high humidity or low humidity gas, the water molecules are exchanged between the environment and the Nafion membrane. At this time, a concentration gradient inside the IPMC is formed, which forces the cations to migrate directionally, thus generating a difference in the cations’ concentration on both sides of the IPMC [4,5,6,7,8,9,10], as shown in Figure 1. In view of the excellent response to water molecules, an IPMC can be potentially used as a voltage humidity sensor for the monitoring of human breathing and water seepage monitoring in a granary. Recently, Wang et al. systematically studied the variation in the capacitance, surface resistance and voltage of a Pd-based IPMC under different humidity gradients, and discussed the influence of IPMC thickness, cation type and immersion reduction electroplating (IRP) times on the humidity sensing performance [11]. In addition, Narimani et al. used the mixture of Nafion and hydroxide to prepare a humidity sensor, and studied the influence of temperature and hydroxide content on the humidity sensing of an IPMC [12].

Many researchers have made great efforts to develop an IPMC humidity sensor with the best sensing performance by far. Emad Esmaeli et al. used Ti and Cr as the adhesion layer between Nafion and Au electrodes to prepare IPMC samples, and its capacitance value was measured under different static humidity environments [8]. Subsequently, Fahimeh Beigi et al. performed an improved study on IPMC humidity sensing. They prepared an IPMC with a Nafion membrane doped with layered double hydroxide (LDH) nanoparticles to increase the water absorption of the IPMC and studied its ionic electrical response to humidity [13]. However, due to the fact that the dense electrode layer of the classical IPMC hinders water molecule exchange between the environment and the IPMC, the classical IPMC is not suitable for humidity sensing [11]. In order to enhance the humidity sensing performance of an IPMC, some functional materials with good conductivity have been used to prepare flexible electrodes on the surface of a Nafion membrane, such as carbon nanotubes, metal nanowires and conductive polymers [14,15,16,17]. Wang et al. deposited a layer of silver nanowire (Ag NW) network on the surface of a Nafion membrane through a spaying process to prepare an Ag-NW-based IPMC humidity sensor [18]. Compared with the noble metal electrodes, these flexible electrodes not only have excellent conductivity and mechanical flexibility but also have a sparse gap structure to promote the water molecule exchange between the environment and the IPMC. However, an Ag-NW-based IPMC is damaged easily due to the low adhesion between the NW’s electrode and the Nafion membrane. To enhance the mechanical stability of functional material electrodes, polymer adhesives were used. Zhang et al. developed a strategy to fabricate flexible electrodes with enhanced reliability and robustness by fully embedding Cu NWs into the surface layer of poly(dimethylsiloxane) (PDMS), followed with a high-intensity pulsed light technique [19]. However, non-conductive polymer adhesives are usually wrapped on the surface of NWs, greatly increasing the surface resistance of electrodes and hindering the output of sensing signals [20,21]. Therefore, a reliable and facile method is urgently needed for preparing a high-performance IPMC with high stability, low sheet resistance and a sparse gap structure to improve the humidity sensing performance [22,23,24].

In this work, we proposed a facile and efficient method to prepare a high-performance IPMC based on Au-shell-Ag-NW-embedded electrodes, which can clearly perceive the change in environmental humidity by directly measuring the electrical signal generated by it. Compared with a traditional Pd-based IPMC and Ag-NW-based IPMC, the Au-shell-Ag-NW-based IPMC has good stability, low sheet resistance and a sparse gap structure, which can effectively enhance the water molecule exchange between the environment and the IPMC, and improve the humidity sensing performance. The rest of our work is arranged as follows. In Section 2, we describe the preparation process of the Au-shell-Ag-NW-based IPMC. In Section 3, the characterization of the Au-shell-Ag NW-based IPMC is described in detail. Secondly, the influence of the parameters in the preparation process on the performance of the Au-shell-Ag-NW-based IPMC was investigated, including the solvent type and electrodepositing time. In Section 4, the paper discusses the excellent humidity sensing performance of the Au-shell-Ag-NW-based IPMC and its application in human breathing monitoring.

## 2. Experimental Section

### 2.1. Material Preparation

In our experiments, Nafion^®^117 membrane (thickness: 183 μm) was selected as the middle layer of IPMC, which was purchased from Dupont ™ (Wilmington, DE, USA). Nafion solution (concentration: 20%) was used to prepare the transition layer between Nafion membrane and electrode layer, which was purchased from Dupont ™ (Wilmington, DE, USA). Ag NWs (diameter: 100 nm, the length range: 100–200 um, concentration: 20 mgmL^−1^, solvent: ethanol) were used to prepare conductive electrodes, and were purchased from XFNANO™ (Nanjing, China). EtOH (99.5%), N, N-dimethylacetamide (DMAC) (99%) were selected as the diluent of Nafion solution and dispersion of Ag NWs, respectively, and were purchased from Aladdin ™ (Shanghai, China). VHB adhesive tape (3MF9460PC) was used to conduct peeling test cycle, and was purchased from 3M™ (Saint Paul, MN, USA). All the reagents were used as received without further purification.

### 2.2. Au-Shell-Ag-NW-Based IPMC Preparation

Different from the traditional IRP process (immersion reduction electroplating based on chemical reaction), this work integrated the spraying and electrodepositing Au process to rapidly prepare IPMC with excellent performance. As shown in Appendix A, the detailed preparation process is described as follows:

(a)Solution preparation. Nafion solution and DMAC were mixed at a ratio of 1:5. Then, magnetic stirring was used to fully dilute Nafion solution in DMAC for obtaining Nafion/DMAC solution. In addition, Ag NW dispersion solution and EtOH were mixed at a ratio of 1:50. Then, Ag NWs were completely dispersed in EtOH solution by ultrasonic vibration to obtain Ag NWs/EtOH solution.(b)Pretreatment of Nafion membrane. The first step was to cut Nafion membrane into 3.5 cm × 3.5 cm, then use sandblasting machine to roughen the surface of Nafion membrane for promoting Nafion/DMAC solution to penetrate into the interior of Nafion membrane and form a stable combination after solidification. The second step was to remove impurities from Nafion membrane. First of all, the quartz sand on the surface of Nafion membrane was removed by ultrasonic cleaning in DI water (30 min, 50 °C). Then Nafion membrane was put into hydrochloric acid solution (HCl, 0.2 mol/L, 100 mL) and DI water in turn, then heated in a water bath (100 °C) for 2 h. The third step was to conduct cation exchange. In this step, Nafion membrane was soaked in 0.2 mol/L NaOH solution for 2 h, and finally put in DI water for standby use.(c)Transition layer preparation. First, Nafion membrane was placed on the glass plate, and two glass plates used to compress the two ends of membrane to prevent it from swelling and deforming after absorbing the Nafion/DMAC solution. Additionally, the temperature of heating platform was set to 120 °C to quickly evaporate DMAC solution. Then, the spray gun was used to uniformly disperse Nafion/DMAC solution on the surface of Nafion membrane (the amount of Nafion/DMAC solution: 0.5 mL/cm^2^). After thermal curing, DMAC and other solvents had evaporated, and transition layer changed from liquid to solid.(d)Au-shell-Ag-NW-embedded electrode preparation. Ag NWs/EtOH solution (1.2 mL) was uniformly dispersed on the surface of the transition layer through the spray gun. Additionally, the temperature of heating platform was set to 90 °C to evaporate EtOH solution. When EtOH had evaporated, Au-shell-Ag-NW-embedded electrodes had been formed. Steps (c) and (d) were repeated to form an electrode layer on the other side. Next, the IPMC was put into the electroplating set-up for electroplating Au for 60 s under 3.5 V. For details of the electroplating set-up refer to our previous work [25].

### 2.3. Characterization

The microstructure of Au-shell-Ag-NW-based IPMC was characterized using a ZEISS Sigma 500 field emission scanning electron microscope (Oberkochen, Germany. The resistance of the sample was measured by a Jingge Electronics Co., Ltd. M3 four-probe meter (Suzhou, China). The humidity sensing performance of IPMC was tested using a self-made humidity sensing platform, which was able to generate a variety of humidity levels (Appendix A). Meantime, a signal acquisition card (NI USB-6001, USA) was used to record the changes in the sensor voltage signal in real time.

## 3. Results and Discussions

### 3.1. Characterization of Au-Shell-Ag-NW-Based IPMC

Figure 2a shows the micromorphology of classical Nafion^®^ IPMC made via the electricity plating of Pd. It can be clearly seen that Pd particles are densely distributed on the surface of Nafion membrane without any gaps, which seriously hinders water molecule exchange between the environment and the Nafion membrane [11]. Figure 2b shows the micromorphology of the pure Ag NW electrode without Au plating and a transition layer. As described in Section 1, a large number of gaps are distributed on the surface of the Nafion membrane, which help the Nafion membrane to quickly absorb or volatilize water molecules. However, Ag NWs are loosely stacked on the surface of the Nafion membrane, which results in the electrode shedding when scratched [17]. Figure 2c shows the cross-section of the Au-shell-Ag-NW-based IPMC. It can be clearly observed that there is no apparent delamination between the transition layer and the Nafion membrane. This is because the Nafion membrane can absorb the Nafion/DMAC solution due to its good moisture absorption property. In addition, DMAC is an organic solution with a high boiling point that can keep liquid at a high temperature for a long time, which provides enough time for the immersion of the Nafion/DMAC solution. When DMAC is evaporated, the immersed Nafion/DMAC solution solidifies, which helps the formation of a seamless connection between the transition layer and the Nafion membrane. Figure 2d shows the surface micromorphology of the Au-shell-Ag-NW-based IPMC. The transparent part is the transition layer. It can be observed that one end of the Ag NWs is firmly embedded in the transition layer, which is used to fix the Ag NWs and collect ionic charges. The other ends of the Ag NWs are exposed on the surface of the transition layer and overlapped with each other, which is used to transmit the sensing signal. This phenomenon is consistent with the preparation principle in Section 2.2. Meanwhile, a large number of gaps are distributed among the Ag NWs, which helps the Nafion membrane to quickly absorb or volatilize water molecules. In order to further demonstrate the microstructure of the Au-shell-Ag-NW-based IPMC, the enlarged SEM image of Figure 2d shows that the smooth and continuous edge of the Ag NWs disappears at the Ag NW–Ag NW junctions, forming a “weld beading” with a relatively large diameter, which indicates that Au atoms fill the gap among the Ag NWs to achieve a sturdy welding connection. Due to the spraying method, we can only macroscopically control the uniformity of the Ag NWs on the surface of the Nafion membrane. However, the microscopic uniformity of the Ag NWs is difficult to achieve. To verify the strength of the Au-shell-Ag-NW-based IPMC, we used a VHB adhesive tape to conduct a peeling test cycle, as shown in Figure 3e. After 50 cyclic sticking attempts with tape, the surface electrode of the Au shell Ag NWs did not peel off and the increase in the sheet resistance of the IPMC was still negligible, which is mainly attributed to the welding connection among the Ag NWs and the firm transition layer between the electrode layer and the Nafion membrane. 

The type and boiling point of the solvent has an important impact on the performance of the Au-shell-Ag-NW-based IPMC. The solvent with a high boiling point has good thermal stability and can keep a liquid state for a long time, which provides enough time to form the electrode layer. The solvent with a low boiling point is highly volatile, which shortens the thermal curing process. In addition, the experiment results show that the strength of the Au-shell-Ag-NW-based IPMC is also related to the type of solvent. To explore the influence of the type of solvent and its boiling point on the performance of the Au-shell-Ag-NW-based IPMC, we used EtOH (organic solvent with low boiling point), DI water (inorganic solvent with low boiling point) and DMAC (organic solvent with high boiling point) as the diluent of the Nafion solution and for the dispersion of the Ag NWs, respectively. As shown in Figure 3, for convenience, the samples are labeled with Xi (X = A, B, C, D and i = 1, 2, 3), and the letters A–D represent, respectively, the Nafion/DMAC solution and Ag NWs/EtOH solution combination, the Nafion/DI solution and Ag NWs/EtOH solution combination, the Nafion/EtOH solution and Ag NWs/EtOH solution combination, and the Nafion/DMAC solution and Ag NWs/DI solution combination. Numbers 1–3 represent the samples after spraying the Nafion mixed solution, Ag NWs dispersion solution and electroplating Au, respectively.

Figure 3 shows the surface of the Au-shell-Ag-NW-based IPMC prepared through different solvents. By comparing samples A1–C1 and A2–C2, it can be clearly seen that the integrity and uniformity of the transition layer and the electrode layer gradually deteriorate with the reduction in the boiling point of the solvent. A large number of cracks appear in B2, and a large area of peeling off occurs in C1 and C2. In contrast, samples A1 and A2 have good integrity and uniformity. This is because the strength of the transition layer mainly depends on the immersion effects between the transition layer and the Nafion membrane. As DMAC is a high boiling point organic solvent, it can keep a liquid state for a long time. When the Nafion/DMAC solution is sprayed on the surface of the Nafion membrane, the solution can maintain a certain fluidity and permeability for 42 s, which provides enough time for the solution to immerse into the Nafion membrane. However, water or EtOH are easy to volatilize, which causes the transition layer to solidify and become brittle quickly. The immersion layer cannot be formed. Therefore, there is an apparent delamination between the transition layer and the Nafion membrane. In addition, the Nafion membrane swells and deforms due to the absorption of EtOH or water, resulting in uneven stress, which causes the electrode to crack due to high brittleness.

In order to further interpret the above phenomena and conclusions, we tested the solidification time of different Nafion mixed solutions and the bending stiffness of the related IPMC. In the experiment, 0.02 mL of Nafion mixed solution was sprayed on the surface of the Nafion membrane (2 cm^2^). Then the solidification time and bending stiffness of IPMC were recorded, as shown in Table 1. It was found that the solidification time of the Nafion/DMAC solution is up to 42 s, while that of the Nafion/EtOH solution is only 2 s. In addition, the bending stiffness of the IPMC increases with the reduction in the solvent boiling point. The bending stiffness of the IPMC using the Nafion/EtOH solution is as high as 482 MPa, which is 2.6 times that of the IPMC using the Nafion/DMAC solution. This is because it is easy for the transition layer to completely solidify and become brittle due to the low boiling point of the solution. However, DMAC can exist inside the transition layer for a long time due to its high boiling point, which makes the transition layer remain soft for a long time to eliminate internal stress. In addition, by observing samples A3 and D3, it was found that the surface of sample D3 peeled off locally, while sample A3 remained intact. The reason is that the solidified Nafion cannot be melted under the action of DI water, so the Ag-NWembedded layer on the surface of the transition layer cannot be formed.

Figure 4a shows the cross-section of sample C3. It was found that when EtOH is used as the diluent of the Nafion solution, there is an apparent delamination at the junction of the Nafion membrane and the transition layer, and the immersion layer is not formed, which further indicates that the low boiling point solvent cannot enhance the strength of the electrodes. Figure 4b shows the cross-section of sample D3. It shows that when water is used for the dispersion of Ag NWs, there is an apparent delamination between the Ag NWs’ conductive layer and the transition layer, which indicates that the Ag-NW-embedded layer cannot be formed under the action of water.

The electrodepositing time is also one of the key factors to determine the performance of the Au-shell-Ag-NW-based IPMC. Figure 5a shows the surface micromorphology of the Au-shell-Ag-NW-based IPMC with different electrodepositing times, including 30 s, 60 s and 80 s. It can be clearly seen that when the electrodepositing time is 30 s, only a small amount of the Ag NWs are electroplated. The Au shell only appears at the tip of the Ag NWs, and the thickness is small. When the electrodepositing time is 60 s, compared with the former, almost all the Ag NWs exposed on the surface are electroplated, and the Au shell is evenly distributed over the entire Ag NWs rather than just the tips. In addition, the weld beading is formed at the Ag NW–Ag NW junctions, indicating that Au atoms fill the gaps among the Ag NWs. In contrast, when the electroplating time is 80 s, the coverage area of the Au shell over the Ag NWs does not increase, but the thickness of the Au shell increases significantly. Meanwhile, the gaps among the Ag NWs are covered by the thickened Au shell, which hinders the Nafion membrane from quickly absorbing or volatilizing water molecules. Figure 5b shows the surface resistance of the Au-shell-Ag-NW-based IPMC with different electrodepositing times, which further proves the above conclusion. It can be found that the surface resistance decreases rapidly in the first 30 s. From 30 s to 60 s, the change rate of surface resistance decreases slowly. The change can be almost ignored after 60 s. The reason is that when the electrodepositing time is 60 s, the Au shell has been completely formed, and almost all the gaps at the Ag NW–Ag NW junctions are welded by Au atoms. Therefore, the conductive path of the electrode layer has been fully formed. However, after 60 s, only the thickness of the Au shell increases, but the effective point contact at the Ag NW–Ag NW junctions does not change.

### 3.2. Humidity Ionic Electrical Response

To implement the humidity sensing test, we established a measurement platform, which can be seen in our previous work [11,18]. According to the humidity sensing mechanism of the IPMC in Section 1, we measured the humidity electrical response of the Au-shell-Ag-NW-based IPMC from 57% RH to 100% RH (humidification process) and from 57% RH to 0% RH (dehumidification process). After the sample is placed in the initial humidity environment (57% RH) for 1 h, the gas with the higher humidity or lower humidity is filled into the chamber for water molecule exchange, then the change in voltage is recorded, as shown in Figure 6. During humidification, the humidity electrical response voltage shows an upward trend, while during dehumidification, the humidity electrical response voltage shows a downward trend. This is because when the humidity level switches from 57% RH to a higher humidity level, the ion concentration on the moisture absorption side of the IPMC decreases due to water absorption. Under the action of the concentration gradient, the cations move to the moisture absorption surface, and vice versa. Moreover, when the humidity changes from 57% RH to 100% RH, the voltage changes by as much as 9.6 mV. During the dehumidification process, the voltage continuously decreases from 0 mV at 57% RH to −4.6 mV at 0% RH. It can be seen that the humidifying process has a higher response speed and response amplitude than the dehumidifying process. This is because Nafion is a hydrophilic material, which can absorb water quickly but dehydrates slowly. Additionally, it should be noted that the humidity electrical response is reversible, which is different from other sensing mechanisms.

In order to further explore the humidity electrical response performance of the Au-shell-Ag-NW-based IPMC, the samples based on different preparation processes were tested and compared. There were three samples for each experiment, with the average value taken. As shown in Figure 7a, it can be seen that the response voltage of the Au-shell-Ag-NW-based IPMC is at least two orders of magnitude higher than that of the Pd-based IPMC in each humidity gradient. Compared with the Ag-NW-based IPMC, the Au-shell-Ag-NW-based IPMC also has a good humidity sensing performance. The reason is that the Pd-based IPMC has a dense surface, which hinders the Nafion membrane from quickly absorbing or volatilizing water molecules, making the humidity response voltage extremely small and even negligible. Although the Ag-NW-based IPMC has a sparse gap structure to enhance the exchange of water molecules, a weak combination between the electrodes and the Nafion membrane leads to the loss of ion charges in the collection and transmission process, leading to the small humidity response voltage. Compared with the former two, the Au-shell-Ag-NW-based IPMC has several advantages to enhance the humidity sensing, including the sparse gap structure for promoting the exchange of water molecules, and the extremely low surface resistance (~3.4 Ohm/sq) for reducing the loss of ion charges in the collection and transmission process. Figure 7b shows the humidity response voltage amplitude (from 57% RH to 100% RH) of the Au-shell-Ag-NW-based IPMC with different electrodepositing times. There are three samples for each experiment. When the electrodepositing time is 30 s, the voltage amplitude at each humidity gradient is low because the high surface resistance hinders the transmission of ionic electrical signals. In addition, when the electrodepositing time is 80 s, the response voltage amplitude drops slightly compared with that at 60 s. This is mainly because when the electrodepositing time is 60 s, the Au shell and conductive path have been completely formed. However, after 60 s, excessive electroplating with Au increases the density of the electrode, which hinders the Nafion membrane from quickly absorbing or volatilizing water molecules. Meanwhile, we characterized the long-term stability by testing the peak voltage of different samples from 57% RH to 100% RH. There were three samples for each experiment, with the average value taken. As shown in Figure 7c, it was found that the Au-shell-Ag-NW-based IPMC and Pd-based IPMC show a high stability within four weeks, while the Ag-NW-based IPMC gradually declines. The reason is that the electrode surfaces of the Au-shell-Ag-NW-based IPMC and the Pd-based IPMC are protected by the Au shell, which can remain stable in air for a long time. However, Ag NWs are easily oxidized or damaged. Comparative table of different methodology is shown in Appendix A.

The Au-shell-Ag-NW-based IPMC has a fast response and high sensitivity to humidity change. Therefore, we explored its application in monitoring the breathing process of humans. We invited a healthy adult male as a volunteer to test his breathing process. Before the breathing test, the volunteer ran quickly for 5 min, then blew air into the chamber, in which the Au-shell-Ag-NW-based IPMC was placed. In our experiment, the relative humidity in the air was about 64% RH, and the humidity electrical response curve of a human’s breathing process is shown in Figure 7d. The humidity electrical response curve is shown as rising first and then falling, which corresponds to the exhalation and inhalation in a human’s breathing process. Unlike the humidity electrical response curve (increasing rapidly at first and then decreasing slowly) in Figure 6, the curve in Figure 7d shows a rapid increase, then a rapid decrease. This is because the air flow around the IPMC is accelerated during the inhalation process, which promotes the rapid dehumidification of the IPMC. In addition, the Au-shell-Ag-NW-based IPMC can accurately detect the human breathing rate, which further indicates its effective application in human health monitoring.

## 4. Conclusions

In this work, we proposed an efficient and rapid method to prepare an Au-shell-Ag-NW-based IPMC with high strength and an excellent humidity sensing performance. Through using the spraying and electroplating Au process, the Au-shell-Ag-NW-based IPMC has several advantages that enhance the humidity sensing, including the sparse gap structure and low sheet resistance (~3.4 Ohm/sq), and the seamless connection at the junction of the Nafion membrane, transition layer and Ag-NW-embedded layer. In addition, the study found that the formation of the transition layer depends on the high boiling point solvent. The formation of the Ag-NW-embedded layer and the Au shell Ag NWs network layer depends on the organic solvent. Compared with a Pd-based IPMC (~0.41 mV), the humidity response voltage (~9.6 mV) of the Au-shell-Ag-NW-based IPMC is at least two orders of magnitude higher in each humidity gradient. Its humidity sensing performance barely changed after being exposed to the air for 4 weeks. In addition, the Au-shell-Ag-NW-based IPMC can effectively monitor the human breathing process, which further indicates that it has good application potential due to its effective preparation method and high stability. However, Nafion solution is expensive, so we will explore cheaper materials to prepare the transition layer of the IPMC in order to reduce the processing costs in the future.

## Figures and Tables

**Figure 1 polymers-15-00733-f001:**
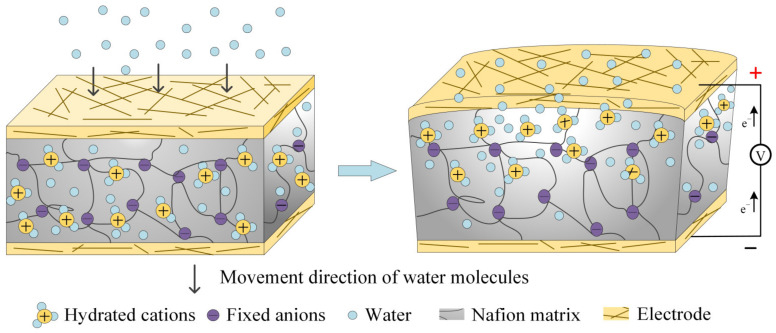
Humidity sensing mechanism of IPMC.

**Figure 2 polymers-15-00733-f002:**
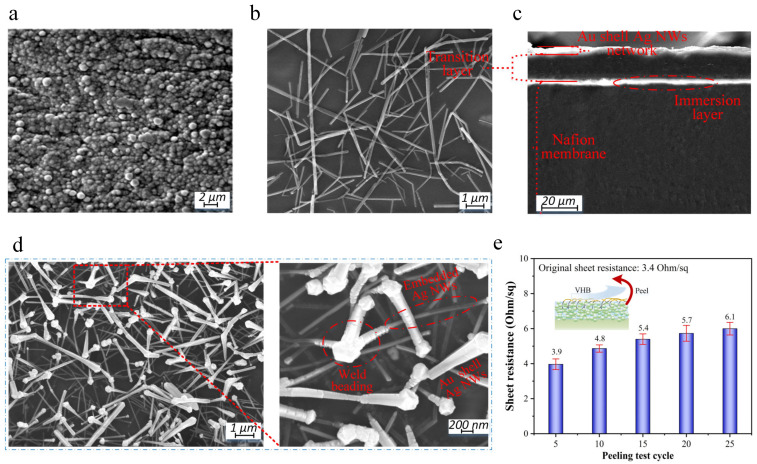
Characterization of different samples. (**a**) Micromorphology of Pd-based IPMC. (**b**) Micromorphology of Ag-NW-based IPMC. (**c**) Cross-section of Au-shell-Ag-NW-based IPMC. (**d**) Micromorphology of Au-shell-Ag-NW-based IPMC. (**e**) Peeling test cycle.

**Figure 3 polymers-15-00733-f003:**
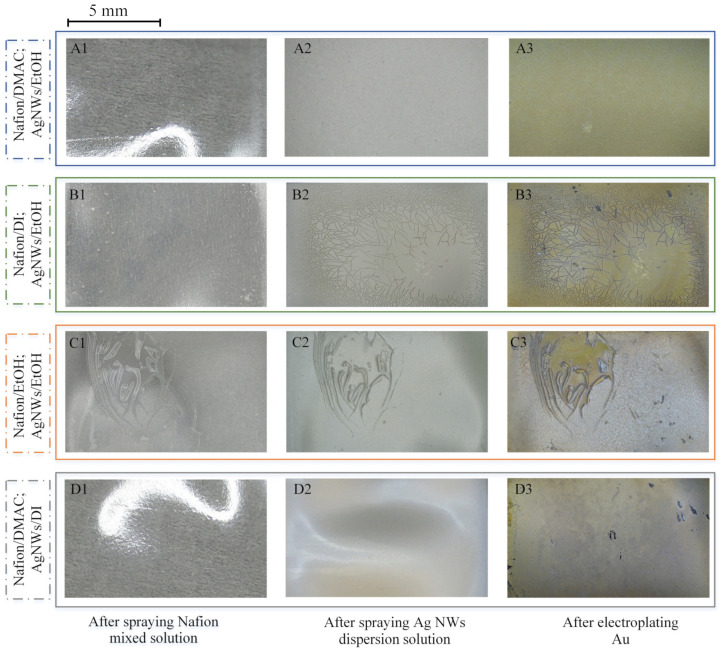
Surface of Au-shell-Ag-NW-based IPMC prepared based on different solvents. The letters A–D represent, respectively, the Nafion/DMAC solution and Ag NWs/EtOH solution combination, the Nafion/DI solution and Ag NWs/EtOH solution combination, the Nafion/EtOH solution and Ag NWs/EtOH solution combination, and the Nafion/DMAC solution and Ag NWs/DI solution combination. Numbers 1–3 represent the samples after spraying the Nafion mixed solution, Ag NWs dispersion solution and electroplating Au, respectively.

**Figure 4 polymers-15-00733-f004:**
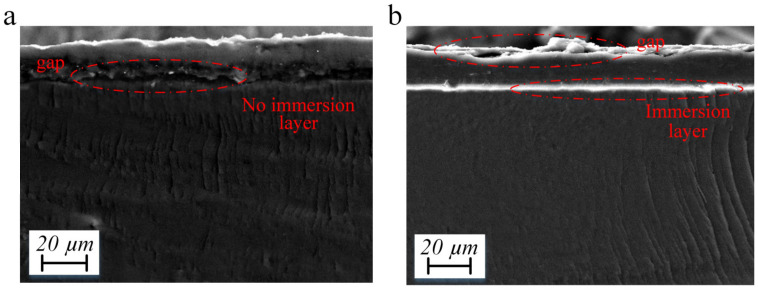
Characterization. (**a**) Cross-section of sample C3. (**b**) Cross-section of sample D3.

**Figure 5 polymers-15-00733-f005:**
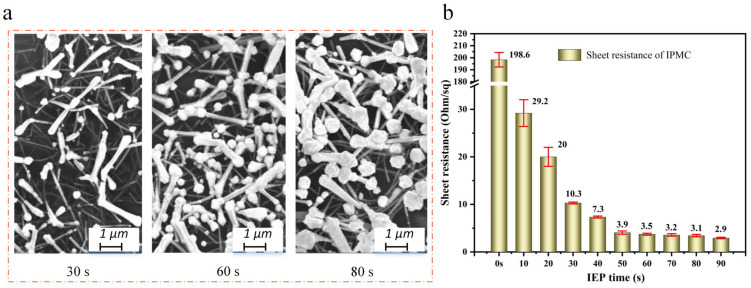
Effects of electroplating time on Au-shell-Ag-NW-based IPMC. (**a**) Micromorphology of IPMC with 30 s, 60 s and 80 s electroplating time, respectively. (**b**) Sheet resistance of IPMC with different electroplating times.

**Figure 6 polymers-15-00733-f006:**
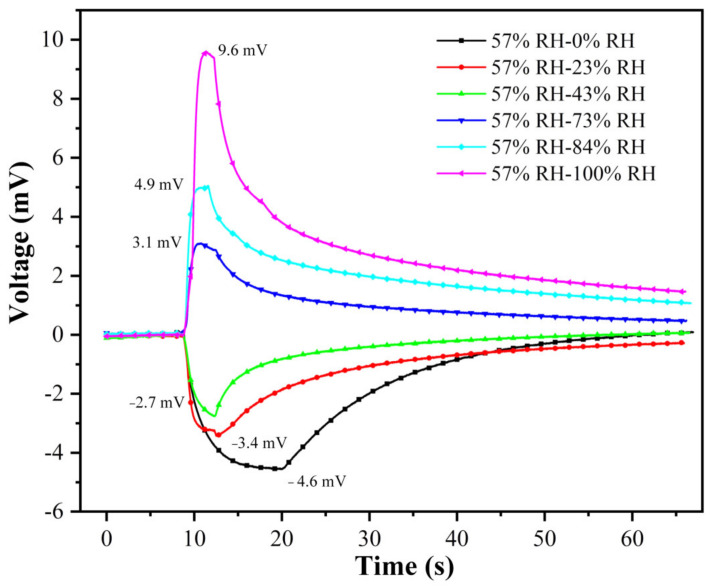
Humidity ionic electrical response of Au-shell-Ag-NW-based IPMC from 57% RH to other humidity levels.

**Figure 7 polymers-15-00733-f007:**
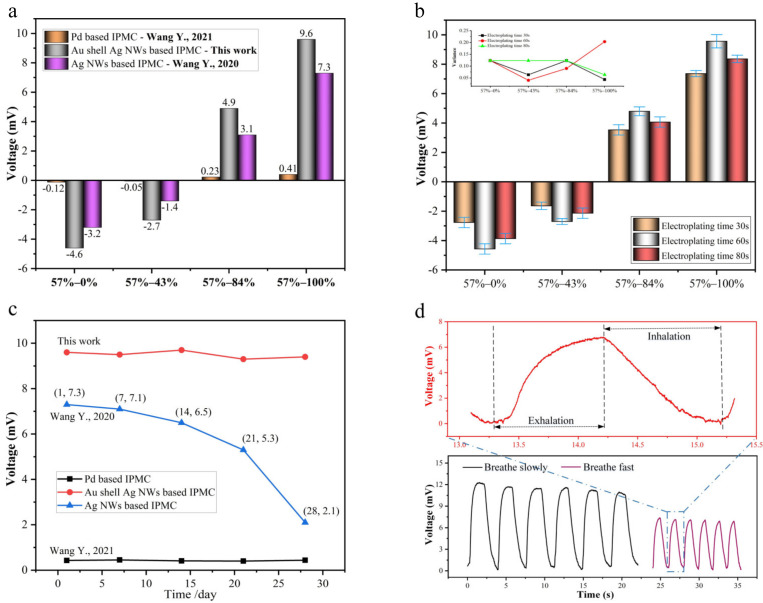
Voltage response of different samples from 57% RH to 100% RH. (**a**) Samples based on different preparation processes [11,18]. (**b**) Samples based on different electrodepositing times. (**c**) Long-term stability of the peak voltage [11,18]. (**d**) Breathing test using Au-shell-Ag-NW-based IPMC.

**Table 1 polymers-15-00733-t001:** Solidification time and bending stiffness of IPMC with different Nafion mixed solutions.

Solution for Preparing Transition Layer	Solidification Time/s	Bending Stiffness/MPa
Nafion/DMAC solution	~42	181
Nafion/DI solution	~6	326
Nafion/EtOH solution	~2	482

## Data Availability

The data presented in this study are available on request from the corresponding author.

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
