# Peer review of "Rapid Preparation of Novel Ionic Polymer–Metal Composite for Improving Humidity Sensing Effect"

_polymers, 2023, doi:10.3390/polym15030733_

Round 1

Reviewer 1 Report

Interesting results and novelty work. A paper focuses on Rapid Preparation of Novel Ionic Polymer Metal Composite for Improving Humidity Sensing Effect. Though the intention of the authors is highly commendable, there is lot of problems particularly in the presentation throughout the manuscript. Besides there are many grammatical mistakes throughout the manuscript, particularly in respect of use of singular and plural with the subject or verb. In view of the above comments, whole manuscript should be properly written to make it acceptable by Polymers journal. I highly recommended this article to be accepted and published in the revised version.

 Abstract:

The abstract given here starts with good background for the present work. Of course, it contains brief details about experimental aspects and the obtained results. However this abstract does not follow the norm of an abstract, which should state briefly:

1.     The purpose of the study undertaken, what are you trying to solve

2.     brief mention of experimental aspects (without using abbreviations)

3.     highlights of the results numerically

4.     Important conclusions based on the obtained results

5.     Potential applications

Therefore, it is suggested that the Abstract to be modified as per the suggestions given above.

 Introduction

Introduction section is long with a many references based on the literature survey conducted by the authors. This is very good. However, this lacks in proper presentation of literature survey, which should have been systematic whereby existing scientific gaps should have been brought out. This should have given justification for the present study, which should be followed by the objectives of this study. In fact there is large amount of literature available on the characterization of Ionic polymer metal composites (IPMC). Similarly, a large number of methods to obtain these materials have been used mentioning their advantages and limitations. None of these have been brought out in this study whereby the authors have not justified why they have chosen the method they have used in their study.  It should be noted that normally 'Introduction' should give brief background through literature survey for the study citing previous published work where-by scientific gaps that exist should be brought out. This would have led to justification for the present study.  It is therefore suggested that ‘Introduction Section’ should be revised as suggested above because this Section is an important one from the point of view of taking up the present study.

In my opinion the paper will have good merit if such applications can be demonstrated and reported. Can you give some example?

Materials and Methods:

Normally, this section should have two main subsections. The first one is Materials which should give details of all materials used in the study, where from they were procured, known characteristics, if available (for e.g. EtOH solution, NaOH, etc. where do you get it, what is the purity of the chemical and etc.).

The second subsection should be Methods, where methodologies used in the study should be given in a systematic way using sub section with numbers for each of the properties. First the processing or preparation aspects of the final material should be given followed by the characterization of prepared materials including preparation of samples for any specific property or morphology studies should be presented in a systematic way. Here one should also clearly mention the number of samples used, any standards followed for variety of properties, make and model of the instruments used for characterization, their accuracy and experimental conditions used, etc.

Better to draw the method you used as Figure 1. This could be clearer to readers.

It should be known to the authors when one publishes any scientific paper, the results presented therein should be such they should be reproducible by any other person when the experiment is repeated using the same materials. In the present paper, it would be difficult for any other person to repeat the experiments because the chosen materials do not have any pre-history, which is required for other researchers to carryout experiments to check the possible reproducibility of the procedure adopted by these authors. This one seem to be missing in this manuscript and might be rejected. Please include this in your manuscript.

Some of the paragraph should be under results and discussion and if it is already there then this becomes repetition and hence can be deleted. Secondly, this Section is methods and hence only results should be mentioned and then it should be discussed preferably comparing it with earlier reported similar results by other researchers.

Results & Discussion

Well written and easy for the reader to understand what the authors have conveyed.

Some of the paragraph should be under Methods and if it is already there then this becomes repetition and hence can be deleted. Secondly, this Section is Results & Discussion and hence only results should be mentioned and then it should be discussed preferably comparing it with earlier reported similar results by other researchers. Please improve this section.

Throughout the manuscript, there are no comparison had been done with other published journal. Therefore, please support your statements with other researcher’s work in the section result and discussion. It should be discussed preferably comparing it with earlier reported similar results by other researchers.

Figure 2 (e) should include the standard deviation.

What are you trying to show in Figure 3. Please do label it and put scale. It is 100 m, 1 km scale? Figure must be clear with the scale. Besides, some of the image is clear such as A2.

Please label Figure 5.

How many sample did for each experiment? Please do ANNOVA test and standard deviation for all data collected and presented.

Conclusions

Conclusions given here are do not reflect what had been achieved including many speculations. It is too long and should be in 1 paragraph. Hence these need to be suitably modified. It may be remembered that this Section forms a summary of all the major observations/ results obtained. Accordingly, here presentation should consist of the main Results or the observations of the study in short sentences probably with bullet points. This should stand alone or form a subsection of a Discussion or Results Section. Hence better to rewrite this Section based on the comments given in the whole text.

General Comments:

The paper though contains some interesting results and novelty work, it lacks in its proper presentation in the whole manuscript. Of course there is need for better language throughout the manuscript. It is suggested that the authors should take the help of native English speaking person to take care of this problem. In view of these, the paper is highly recommended and should be accepted for publication in the revised form. It is suggested that the authors should revise the paper in the light of above comments/suggestions.

Reviewer 2 Report

First of all, I would like to thank the authors for the work done. I consider that the idea that is proposed is very interesting. However, there are some things that must be clarified before the work can be published.

My comments are the following:

Abstract:

- NWs are mentioned but what they are is not explained. The first time an acronym appears, its meaning must be specified to make reading and understanding easier for the person who is reading it.

Introduction:

- In line 3 of this section, you must remove the comma that goes before the "and" so that the sentence makes sense.

- In Figure 1, it should be indicated what is happening in the arrow.

Material preparation:

- 20 mg mL-1: the value of -1 must go as a superscript.

- In section 2.2, what do the authors mean when they say "chemical method"? It should be explained in the text and name the differences that exist between the chemical method and the method proposed in the manuscript.

- section 2.2 a): Why is the proportion of 1:50 used? Has it been taken from any previous work or have the authors of this work optimized it? How do you know that AgNWs are completely dispersed in EtOH by ultrasonication? Please explain it.

- section 2.2 b): 3.5x3.5 cm; 30 min; 100mL. Just like there are spaces between numbers and units, in this case it should be exactly the same. This occurs many times throughout the text. Please, always use the same criteria and respect the rules dictated by the magazine. correct it.

To remove impurities, they do it in water and using ultrasound for 30 min at 50ºC and then add HCl. Is HCl also made ultrasonically? If not, how do they do it? explain it.

In this section, abbreviated and non-abbreviated deionized water is named. Always use the same criteria.

- Section 2.2 c): 0.5 mL cm-2: the value of -2 must go as a superscript.

- Section 2.2 d): Literature 24 should not be put like that. The reference is indicated.

I imagine that at the end of step d), c) and d) are repeated again as the authors indicate, but does the system look good like this? Has the effectiveness and reproducibility of the proposed methodology been studied?

Results and Discussion:

- Indicate what VHB adhesive means.

In Table 1, the authors indicate in which units the solidification time values are given in parentheses. However, in the flexural rigidity, the value of the units of the results obtained is not indicated. The ideal would be to put the units in parentheses in the title and only indicate the values obtained in the table.

In Figure 4, it must be specified that sample C3 appears corresponds to figure a) and sample D3 corresponds to figure b).

What is the magnification of the 5a image for 60 and 80 seconds?

Humidity:

- It does not put "Reference 11 and 17". Both are referenced.

Correct "1hours". It's 1 hour.

There is an error in the units ~3.4 Ω/â–¡

Review the English of the conclusions. There are sentences that do not make sense and the word "and" is abused, making the text meaningless.

Have interference studies been done?

Can the material be recycled?

What are the drawbacks of the methodology proposed in this work? Indicate them in the text.

Include a comparative table of this methodology against those that are normally used, indicating the most relevant parameters, the concentrations used, the advantages and disadvantages, the development of the method (if it is simple, the steps used, if toxic solvents are used, etc.). etc)

Reviewer 3 Report

The submitted manuscript with the title “Rapid Preparation of Novel Ionic Polymer Metal Composite for Improving Humidity Sensing Effect” presents an interesting experimental study to improve the sensitivity to humidity using an ionic polymer metal composite. Overall, the manuscript is well organized and clear with high scientific quality.

 I have few comments/suggestions to be considered before publication:

 In Figure 2 the red text in some figures is hard to read.

·        The designation Au shell Ag NWs is not common.

·        Figure 2e should contain more detail. For example, a 3D figure showing the composition and format of the Au shell Ag NWs would be helpful to understand even to complement figure 5.

·        The fabrication of the Au shell Ag NWs is not clear and should be improved.

·        Line 216 Nafon - Nafion

·        Line 310 correct resistance units.

·        The section 4 should be included in section 3 Results and discussions.

·        The information about the setup used for the humidity measurements are missing. The authors pointed to previous publications but is important to include it in this manuscript.

·        In Figure 6 is not clear. The voltage reaches a maximum or a minimum corresponding to 0, 43, 84 and 100%RH but what happens after that point?

·        A reproducibility study should be presented using the humidity chamber with different levels of RH.

·        Figure 7c shows very important results however some error bars are missing and should be included. The same in figures 7a and 7b. This type of graph is not ideal in my opinion, but it is important to include error bars to enhance the improvement or not when using Au shell Ag NWs based IPMC.

·        The results in figure 7d should be explored. It is important to study the response time to RH and for example using different breathing frequencies.

·        In the conclusions section the authors declares that this Au shell Ag NWs based IPMC has high stability and good repeatability but from the presented results this is something that is not possible to conclude.
